# Circulatory miR-411-5p as a Novel Prognostic Biomarker for Major Adverse Cardiovascular Events in Patients with Atrial Fibrillation

**DOI:** 10.3390/ijms24043861

**Published:** 2023-02-15

**Authors:** Stephan Nopp, M. Leontien van der Bent, Daniel Kraemmer, Oliver Königsbrügge, Johann Wojta, Ingrid Pabinger, Cihan Ay, Anne Yaël Nossent

**Affiliations:** 1Clinical Division of Haematology and Haemostaseology, Department of Medicine I, Medical University of Vienna, 1090 Vienna, Austria; 2Department of Surgery and Einthoven Laboratory for Experimental Vascular Medicine, Leiden University Medical Center, 2333 Leiden, The Netherlands; 3Department of Internal Medicine II, Medical University of Vienna, 1090 Vienna, Austria; 4Ludwig Boltzmann Institute for Cardiovascular Research, 1090 Vienna, Austria; 5Department of Laboratory Medicine, Medical University of Vienna, 1090 Vienna, Austria

**Keywords:** circulating microRNA, atrial fibrillation, biomarkers, sequence analysis, RNA, humans

## Abstract

The risk stratification of patients with atrial fibrillation (AF) for subsequent cardiovascular events could help in guiding prevention strategies. In this study, we aimed at investigating circulating microRNAs as prognostic biomarkers for major adverse cardiovascular events (MACE) in AF patients. We conducted a three-stage nested case–control study within the framework of a prospective registry, including 347 AF patients. First, total small RNA-sequencing was performed in 26 patients (13 cases with MACE) and the differential expression of microRNAs was analyzed. Seven candidate microRNAs with promising results in a subgroup analysis on cardiovascular death were selected and measured via using RT-qPCR in 97 patients (42 cases with cardiovascular death). To further validate our findings and investigate broader clinical applicability, we analyzed the same microRNAs in a subsequent nested case–control study of 102 patients (37 cases with early MACE) by using Cox regression. In the microRNA discovery cohort (n = 26), we detected 184 well-expressed microRNAs in circulation without overt differential expression between the cases and controls. A subgroup analysis on cardiovascular death revealed 26 microRNAs that were differentially expressed at a significance level < 0.05 (three of which with an FDR-adjusted *p*-value <0.05). We, therefore, proceeded with a nested case–control approach (n = 97) focusing on patients with cardiovascular death and selected, in total, seven microRNAs for further RT-qPCR analysis. One microRNA, miR-411-5p, was significantly associated with cardiovascular death (adjusted HR (95% CI): 1.95 (1.04–3.67)). Further validation (n = 102) in patients who developed early MACE showed similar results (adjusted HR (95% CI) 2.35 (1.17–4.73)). In conclusion, circulating miR-411-5p could be a valuable prognostic biomarker for MACE in AF patients.

## 1. Introduction

Atrial fibrillation (AF), the most common clinical cardiac arrhythmia, is associated with high morbidity and mortality. It is largely attributable to an increased risk of stroke, myocardial infarction, heart failure, and bleeding [1]. As the AF population is highly heterogeneous in terms of underlying risk for unfavorable outcomes, risk stratification is of high clinical importance to mitigate the risks for related complications. However, current clinical prediction scores show only moderate predictive performance for the respective outcomes, as they are often solely based on clinical parameters [2]. In contrast, biomarkers can reflect the entire spectrum of disease, improve the biological understanding of underlying pathophysiological mechanisms, and may, therefore, provide a more powerful approach to risk prediction. As such, microRNAs have emerged as a special research interest over the past decade.

MicroRNAs are small non-coding RNAs that consist of 18–25 nucleotides. They regulate gene expression at a post-transcriptional level by binding to the 3′ untranslated region of their target messenger RNA (mRNA) through the so-called “seed sequence” on nucleotides 2 to 8. In addition, they inhibit the translation and/or induce the degradation of its target mRNA. As such, microRNAs function as regulators for hundreds of mRNAs; often, at once within a regulatory network [3]. Thus, they substantially affect the pathophysiological pathways that influence disease progression and hold valuable therapeutic potential [4]. Moreover, microRNAs are present and stable in circulation, making them optimal candidate biomarkers. Previous studies have already linked differentially expressed microRNAs with various cardiovascular diseases [5,6,7,8,9,10,11]. Several microRNAs have been identified to be associated with the occurrence or pathological mechanisms of AF [12,13,14,15]. However, to our knowledge, no study has yet prospectively investigated the association of differentially expressed microRNAs with the occurrence of adverse cardiovascular outcomes in AF patients. We, therefore, aimed at identifying the microRNAs associated with future major adverse cardiovascular events (MACE) in patients with AF.

## 2. Results

### 2.1. Study Population

The total study population consisted of 347 patients with AF (mean age: 71.3 years, 37.3% female) and available plasma samples at baseline (Figure 1). Forty-seven MACE occurred in 42 (12.1%) patients during a median in-person follow-up time of 316 (interquartile range [IQR], 233–497) days. Thirty-seven (10.7%) patients had MACE within the first year after study inclusion. A detailed list of events per outcome is available in the Appendix A. Follow-up from the national death registry on the survival status and cause of death was available with a median follow-up time of 4.3 years (IQR, 1.8–5.7). In total, 81 patients died (23.3% of the study cohort); in 42 patients (51.9%), death was attributed to cardiovascular causes, while in 4 (4.9%) patients, the cause of death was not reported.

### 2.2. Selection of Candidate microRNAs in the Discovery Cohort

The results of the differential expression of the microRNAs in the small RNA sequencing are presented in Appendix A. We detected 184 well-expressed microRNAs that were present in circulation of the cases and controls (n = 26). The differences in microRNA expression between the patients who developed three-point (3P)-MACE and their matched controls were not statistically significant when controlling for the false discovery rate (FDR). However, in the subgroup analysis on patients with subsequent cardiovascular death (six cases vs. their matched controls), three microRNAs (miR-150-5p, miR-127-3p, and miR-1908-5p) were significantly associated with cardiovascular mortality (FDR-adjusted *p* < 0.05); an additional 23 microRNAs met the less stringent threshold of *p* < 0.05. We, therefore, decided to proceed with investigating cardiovascular death as the primary endpoint in the subsequent nested case–control study. Of those 26 microRNAs, we selected seven promising microRNAs (miR-150-5p, miR-127-3p, miR-1908-5p, miR-483-3p, miR-411-5p, miR-625-5p, and miR-15a-5p) based on the composite of statistical significance, log fold-change, the visual inspection of the boxplots of the cases vs. controls, and the known biological functions from the cardiovascular literature [16,17,18,19,20,21,22,23] for the purposes of a validation in a nested case–control study with cardiovascular death as the outcome.

### 2.3. Validation of Candidate microRNAs for Cardiovascular Death

The baseline characteristics of the nested case–control cardiovascular death outcome study are presented in Table 1. Note that this analysis was based on the longer follow-up period available from the national death registry (median follow-up time of 4.3 years). The associations of each microRNA with cardiovascular death, based on RT-qPCR measurements, are shown in Figure 2A, whereby the hazard ratios (HRs) are per one standard deviation increase (1-SD). The levels of miR-411-5p, miR-625-5p, and miR-15a-5p had to be dichotomized due to the low detectability (<20% valid values), and the HRs, therefore, compare detectable with undetectable microRNA in a binary fashion. In a multivariable analysis, adjusted for age and sex (model 1), only the miR-411-5p was significantly associated with the outcome (HR: 1.95; 95%CI 1.04–3.67; *p* = 0.038). After a further adjustment with respect to the history of myocardial infarction, as well as to stroke/TIA/systemic embolism (model 2), the detectable miR-411-5p was associated with cardiovascular death by a HR of 1.89 (95%CI 1.00–3.58; *p* = 0.050).

We proceeded to test those findings from the first validation phase using the broader and more clinically relevant cardiovascular endpoint MACE and compared patients who developed MACE within one year of follow-up to the matched controls in a subsequent nested case–control study.

### 2.4. Validation of Candidate microRNAs for Major Adverse Cardiovascular Events

The baseline characteristics of the nested case–control MACE outcome study are presented in Table 2. The associations of each microRNA with MACE are presented in Figure 2B. Notably, this analysis showed similar findings compared to the previous analysis on cardiovascular death. Again, miR-411-5p was significantly associated with the outcome (adjusted HR from model 1: 2.35; 95%CI 1.17–4.73; *p* = 0.017). After adjustment for age, sex, history of myocardial infarction, and stroke/TIA/systemic embolism, the association remained significant (adjusted HR 2.62, 95%CI 1.29–5.33; *p* = 0.008). Eight patients with early cardiovascular death fulfilled the criteria for cases in both validation studies and were, therefore, included in both studies.

Kaplan–Meier curves for both outcome studies were stratified by the detectability of miR-411-5p, and are shown in Figure 3.

## 3. Discussion

In this three-stage nested case–control study, miR-411-5p was significantly associated with adverse cardiovascular events in patients with AF. After the initial microRNA screening phase, we selected seven microRNAs and performed two validation phases with different endpoints and follow-up periods. In both validation phases, AF patients with detectable levels of miR-411-5p at study inclusion were at, approximately, a two-fold increased risk for developing the cardiovascular outcome (i.e., cardiovascular death or MACE).

Previous studies mechanistically linked several microRNAs with pathophysiological pathways for AF and proposed microRNAs as a diagnostic tool for AF [12,13,14,15]. Furthermore, several circulating microRNAs have already been associated with future cardiovascular events, including myocardial infarction [10,24,25] and stroke [5,26]. The present study is the first to focus on the predictive role of microRNAs with regard to future cardiovascular events in patients with AF. We have chosen seven promising microRNAs that were differentially expressed between cases and controls in the discovery phase of our study for further validation. All of the selected microRNAs have been reported in the literature in the context of cardiovascular biology [16,17,18,19,20,21], e.g., miR-150-5p appears to be highly involved in platelet physiology and endothelial cell function [17,27]. In addition, it has also been associated with adverse cardiovascular outcomes [17,22]. Further, miR-1908-5p has been found to play an important role in regulating low-density lipoprotein, total cholesterol, fasting glucose, HbA1c, and several lipid-metabolites in blood [18]. Moreover, miR-15a-5p, which belongs to the miR-15 family, affects angiogenesis and might be a promising biomarker in critical limb ischemia [21,23]. However, only miR-411-5p was significantly associated with cardiovascular endpoints in both the validation phases of our study.

Thus far, the role of miR-411-5p has not been explored in prospective cardiovascular outcome studies, but several preclinical studies and one clinical study have suggested a central role of miR-411-5p in cardiovascular diseases [28,29,30,31,32,33]. MiR-411 is located on chromosome 14 and belongs to the 14q32 microRNA cluster. This genomic region holds one of the largest microRNA clusters, comprising 54 microRNA genes [34] and is known to play a central regulatory role in vascular biology [16,35]. The inhibition of microRNAs from the 14q32 cluster led to increased post-ischemic blood flow recovery and tissue perfusion via neovascularization [29], decreased plasma cholesterol, atherosclerotic lesion formation, and increased plaque stability in mouse models [36,37]. Interestingly, miR-411 also has a 5′ isomiR, which targets a different set of genes than the wildtype miR-411, and is upregulated relative to miR-411 in chronically ischemic human blood vessels [38]. However, in our qPCR analysis, we did not further differentiate between both isoforms. Apart from cardiovascular diseases, miR-411-5p has been attributed a protective role in diabetic retinopathy [39], but most research on miR-411 has been conducted in the field of cancer [40,41,42]. It has been suggested that miR-411 plays a major role in cancer development [43] and the formation of metastases [44,45]. MiR-411 is abnormally expressed in various cancers but with differing roles between tumor types. Several studies have reported on the downregulation of miR-411 and its role as a tumor suppressor in some cancer types, while also reporting on an overexpression and a role in cancer development in others [46]. In the cardiovascular clinical setting, miR-411 has been suggested as a candidate biomarker in only one study. There, miR-411 was significantly upregulated in the whole blood of patients with peripheral artery disease and of patients with abdominal aortic aneurysm, compared to healthy controls [33]. Our study adds to these findings and suggests a potential role for miR-411-5p in the prediction and risk stratification for cardiovascular death and MACE in patients with AF.

As to why miR-411-5p ends up in circulation, where exactly it originates from, and whether its presence in circulation has a role in cell-to-cell communication, is still unknown. From mouse studies performed on the 14q32 locus, the microRNA cluster from which miR-411 is derived, we could extrapolate that miR-411 may be involved in neovascularization and atherosclerotic lesion formation, as well as, importantly, with lesion (in)stability [29,36,37]. However, further preclinical research is needed to evaluate whether elevated circulatory miR-411-5p is causally linked to cardiovascular events or, on the contrary, upregulated due to cardioprotective effects. More clinical research is needed to externally validate our findings and to further investigate the association of miR-411-5p with future cardiovascular events in patients with AF. Potentially, miR-411-5p plasma levels may, ultimately, be used to guide primary or secondary cardiovascular prevention strategies in patients with AF, preferably in the context of a biomarker panel containing several different small RNA biomarkers.

Our study should be viewed in the context of the following limitations: First, microRNA research, in general, has a pronounced problem with replication due to false positive findings. We tried to account for this by controlling the false discovery rate during the screening phase of our study and then proceeded with a small selection of candidate microRNAs in two validation phases. Second, validation phases were not completely independent as eight cases in validation phase II were also included as cases in validation phase I. Third, each of the three nested case–control studies was limited by small sample sizes. Further, we had to analyze three of the seven microRNAs in the validation phases in a dichotomized fashion due to the low detectability of these microRNAs by qPCR. Both factors limit the power of our analysis. This might be the reason for not detecting significant associations with other microRNAs, e.g., miR-15a-5p, even though we expected otherwise based on previous evidence from the literature [21,23]. Nevertheless, we observed significant associations of miR-411-5p with cardiovascular death and MACE. Given the biological plausibility of this finding, as well as with respect to the same direction and magnitude of effect in the discovery phase and both validation phases, miR-411-5p may indeed prove a valuable biomarker for adverse cardiovascular outcomes in patients with AF.

To conclude, miR-411-5p was associated with adverse cardiovascular outcomes in patients with AF in all phases of our three-stage nested case–control study. If miR-411-5p was detectable in the circulation, patients were at approximately two-fold increased risk for cardiovascular death and MACE. Previous studies have shown that miR-411-5p plays a role in various forms of vascular remodeling, which may help explain its implication in cardiovascular death and MACE. However, future studies are needed to externally validate our findings and to further elucidate the precise biological role of miR-411-5p in cardiovascular events.

## 4. Materials and Methods

### 4.1. Study Population and Data Source

Our multistage, nested case–control study was based on a prospective registry that recruited 347 patients with AF, who were referred to the Clinical Division of Hematology and Hemostaseology at the Medical University of Vienna, Austria, between July 2013 and May 2016. Patients with a diagnosis of AF were confirmed by a 12-channel, resting electrocardiogram and the willingness to comply with the study procedures were how the participants were deemed eligible for inclusion; no exclusion criteria were defined. The detailed study procedures have been previously published elsewhere [47]. In brief, a detailed medical history and blood samples were taken at study inclusion. AF patients were then followed for at least 12 months. Follow-up was performed with a particular focus on cardiovascular events, including stroke, transitory ischemic attack (TIA), systemic embolism, myocardial infarction, coronary revascularization, hospitalization for heart failure, and death. Moreover, the national death registry was queried periodically, most recently in May 2020, to identify the deceased patients. Telephone interviews with family members and the review of electronic medical records were used to verify whether the death was due to cardiovascular causes.

This study conforms to the ethical guidelines of the 1975 Declaration of Helsinki. In addition, it was approved by the local ethics committee (number 1665/2012), and participants provided written informed consent.

### 4.2. Study Design

In the present study, we followed a sequential approach starting with screening for potential candidate microRNAs in a discovery case–control study. Subsequently, the most promising microRNAs of the discovery phase were tested in two nested case–control studies.

### 4.3. Discovery Phase

In the first stage, RNA sequencing was performed in a discovery case–control study to identify promising candidate microRNAs in order to predict MACE in patients with AF. For this screening phase, we chose the more stringent 3P-MACE endpoint—defined as stroke, myocardial infarction, or cardiovascular death. Patients who developed 3P-MACE during follow-up were eligible as cases. In total, 13 cases were randomly selected from our AF cohort and matched 1:1 to controls by age (±2 years), sex, and CHA_2_DS_2_-VASc-Score (±1 score point). Notably, the subgroup of patients with cardiovascular death showed the most apparent differential expression pattern compared to their controls. Thus, we first proceeded with investigating the most promising microRNAs in a nested case–control study with cases defined as patients who died from cardiovascular causes during follow-up.

### 4.4. Validation Phase I: Patients with Cardiovascular Death

Seven candidate microRNAs were selected from the discovery phase, based on statistical significance, log fold change, and known biological functions from the cardiovascular literature. Selected microRNAs were measured using reverse-transcription quantitative polymerase chain reaction (RT-qPCR) in all patients with cardiovascular death (n = 42), labeled as cases, as well as their age- and sex-matched controls (n = 55). Matching was performed in a 1:2 ratio with replacement; therefore, the same control could be matched to several cases.

### 4.5. Validation Phase II: Patients with MACE at One Year

In a third step, we aimed to further validate our findings from the cardiovascular death outcome study using a more clinically relevant endpoint, i.e., MACE within the first year of follow-up. Patients who experienced MACE within one year after study inclusion were selected as cases (n = 37). Again, cases were matched in a 1:2 ratio to age- and sex-matched controls (n = 65), allowing for a control to be matched to several cases. MACE was defined as the composite endpoint of stroke, TIA, systemic embolism, myocardial infarction, coronary revascularization, hospitalization for heart failure, and all-cause death.

### 4.6. Laboratory and Statistical Methods

#### 4.6.1. Sample Collection

At study entry, non-fasting venous blood was collected in vacuum tubes containing 3.2% trisodium citrate (Vacuette^®^ Greiner Bio-One, Kremsmünster, Austria). The tubes were centrifuged at 2500 g for 15 min and platelet-poor plasma was isolated. Aliquots with 200 µL each were generated and stored at −80 °C until needed for analysis. All blood samples were processed within the first two hours after blood was drawn.

#### 4.6.2. Discovery Phase: Small RNA Sequencing

For the screening of candidate microRNAs, the total RNA was isolated from 200 µL of citrated plasma using the miRNeasy Serum/Plasma Advanced Kit (Qiagen, Venlo, Netherlands), which was performed according to the manufacturer’s instructions. Next, the small RNAs derived from plasma were sent to BGI (Shenzhen, China) for small RNA sequencing. In brief, BGI small RNA sequencing services were executed using unique molecular identifiers (UMIs), small RNA library construction, and RNA sequencing on the DNBseq Platform. Raw data were de-multiplexed and filtered to remove adapter sequences, contamination, and low-quality reads.

#### 4.6.3. Discovery Phase: Statistical Analysis

The small RNA sequencing dataset was analyzed using the virtual machine of the sRNAtoolbox [48], which is a collection of small RNA research tools that can be used for expression profiling and subsequent downstream analysis. The sRNAbench module was used to deduplicate reads based on UMIs and subsequently map them to the human genome. Several libraries were used for annotation, including miRbase, gtRNAdb, RNAcentral, as well as for NCBI ncRNA and cDNA libraries. Finally, an analysis of differential expression was performed using the sRNAde module. With microRNA abundance normalized to the total number of genome-mapped sequencing reads (expressed as reads per million), the Bioconductor package edgeR was used to investigate differential microRNA expression, while also using the standard quantile-adjusted conditional maximum likelihood method. To account for multiple testing during the discovery phase, we adjusted *p*-values using the method of Benjamini and Hochberg for controlling the FDR.

#### 4.6.4. Validation Phases: Quantification Using RT-qPCR

The seven microRNAs selected during the discovery phase were then measured by RT-qPCR in triplets for each sample and each microRNA using Taqman™ microRNA assays (ThermoFisher, Waltham, MA, USA) (Appendix A). First, the circulating microRNAs were isolated using the miRNeasy Serum/Plasma Kit (Qiagen, Venlo, Netherlands). Second, microRNAs were reverse-transcribed to cDNA with the Taqman™ MicroRNA Reverse Transcription Kit (ThermoFisher, Waltham, MA, USA). Third, samples were measured in triplicate using RT-qPCR on the CFX384 Touch Real-Time PCR Detection System (Bio Rad, Hercules, USA). Cel-miR-39 was used as a spike-in control to normalize the expression levels of microRNAs, using the delta threshold cycle (∆C_T_) method. Three μL of 5 nM cel-miR-39 was added to a total sum of 15 μL of reverse transcription mix [49]. All microRNAs were measured in a blinded fashion, regarding the case vs. control status of the samples.

#### 4.6.5. Validation Phases: Statistical Analysis

Patient characteristics are presented as frequencies (percentage) for the categorical variables, as well as the mean (SD) or median (IQR) for continuous data.

Quantitative PCR results were analyzed according to the data handling pipeline suggested by de Ronde et al. [50]. If the mean quantification cycle (*C*q) of the triplets performed for each microRNA was above a threshold of 38, the microRNA was labeled as undetectable and the value set to 39. Values were then normalized to cel-miR-39 using the ∆C_T_ method. Measures that were labeled as invalid by the data handling pipeline were imputed, through using multiple imputation with chained equations (MICE). Due to the low detectability of some microRNAs, we had to adjust our analysis plan after initial data analysis; the microRNAs with less than 20% of valid values in all participants were dichotomized based on detectability (i.e., miR-625-5p, miR-15a-5p, and miR-411-5p). If the microRNA was measurable in any of the triplets of each sample, it was counted as detectable.

Finally, the associations of microRNAs with the outcomes (i.e., cardiovascular death and MACE) were assessed using Cox proportional hazard models. Multivariable analysis was performed using two models; model 1 was adjusted for age and sex; model 2 included the variables of model 1 and the history of myocardial infarction, as well as the history of stroke/TIA/systemic embolism. For visualization, Kaplan–Maier curves were plotted and stratified by the detectability of the microRNA of interest, as well as were compared using the log-rank test. All statistical tests were two-tailed with an alpha level of 0.05 denoting statistical significance. Data were analyzed in R (Version 4.1.0; R Core Team, 2019) using the MatchIt, mice, and survival package, as well as the R script provided by de Ronde et al. [50].

## Figures and Tables

**Figure 1 ijms-24-03861-f001:**
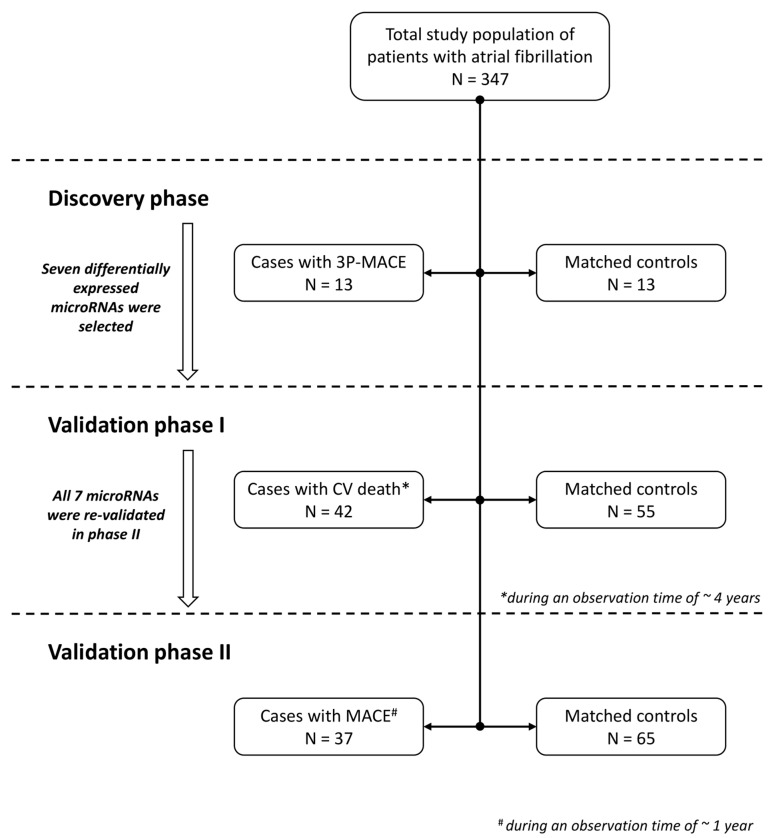
Study flow chart. (Abbreviation: 3P-MACE, three point major adverse cardiovascular event; CV, cardiovascular; and MACE, major adverse cardiovascular event).

**Figure 2 ijms-24-03861-f002:**
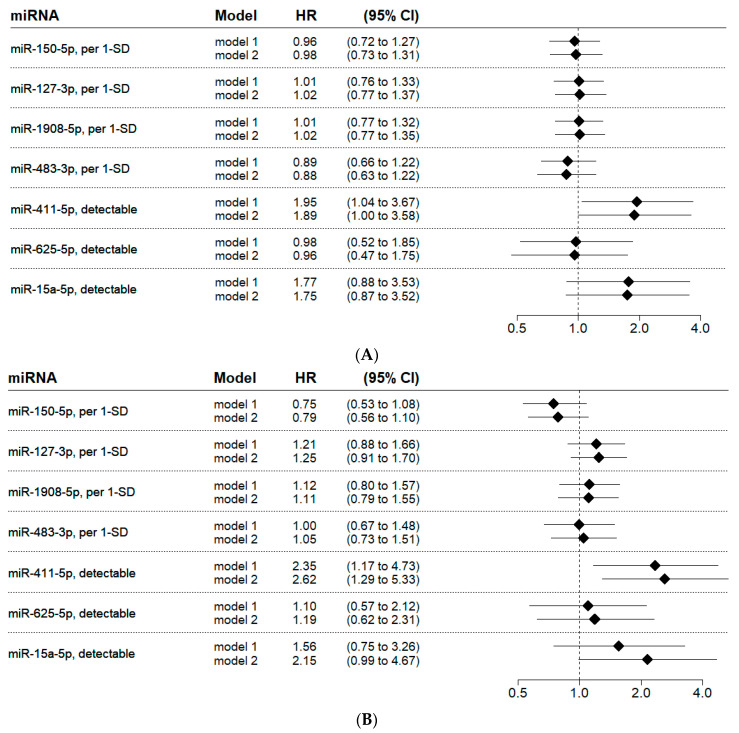
Association of candidate microRNAs with cardiovascular death (**A**) and major adverse cardiovascular events (**B**) in patients with atrial fibrillation. Model 1 includes the adjustment for age and sex. Model 2, in addition to the variables of model 1, further adjusts for the history of myocardial infarction and the history of stroke/TIA/systemic embolism. Due to the low detectability of miR-411-5p, miR-625-5p, and miR-15a-5p in RT-qPCR, the variables were dichotomized based on the detectability of the microRNA. Abbreviations: 1-SD, one standard deviation; CI, confidence interval; HR, hazard ratio; and SD, standard deviation.

**Figure 3 ijms-24-03861-f003:**
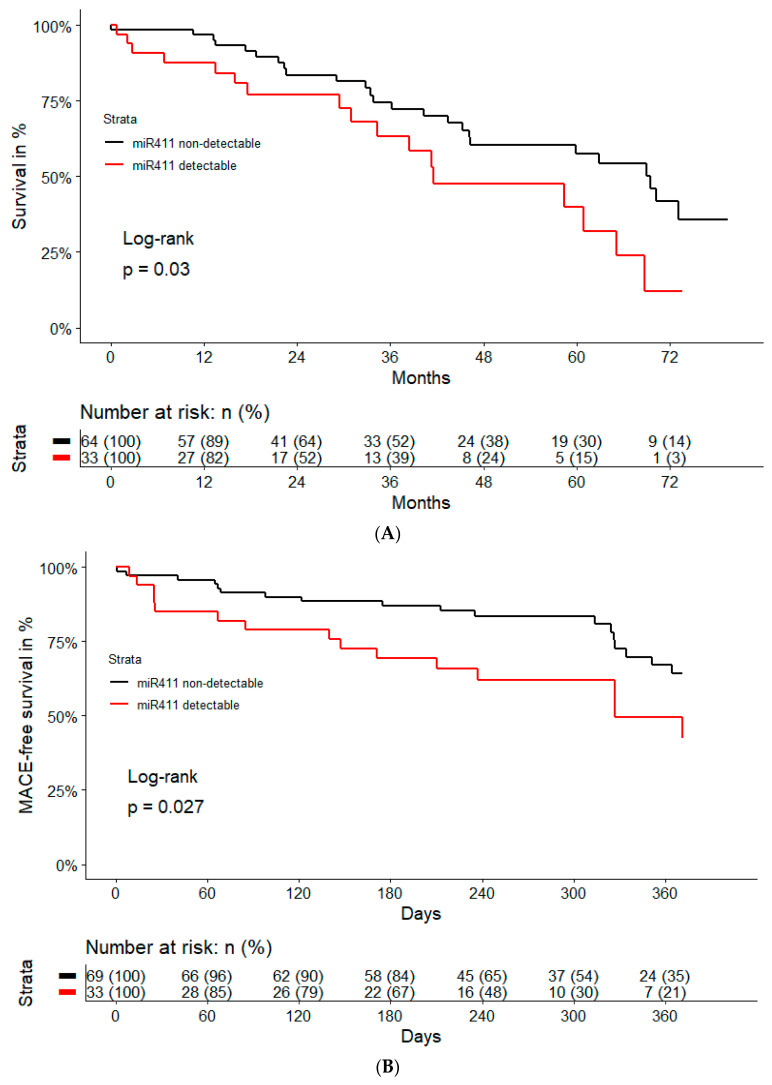
Kaplan–Meier curves for cardiovascular death (**A**) and major adverse cardiovascular events (**B**), as stratified by the detectability of miR-411-5p in patients with atrial fibrillation.

**Table 1 ijms-24-03861-t001:** Baseline characteristics of patients with atrial fibrillation: cases with cardiovascular death vs. matched controls without cardiovascular death.

	Cases with Cardiovascular Death (n = 42)	Controls (n = 55)
Demographics		
Age, years	77.0 (71.3–80.8)	76.0 (71.0–80.0)
Male sex	28 (66.7%)	37 (67.3%)
BMI, kg/m^2^ [2]	26.6 (24.6–29.0)	26.9 (25.3–30.7)
Information on atrial fibrillation, n (%)		
Type of atrial fibrillation [2]		
First onset	4 (9.5%)	6 (10.9%)
Paroxysmal	18 (42.9%)	26 (47.3%)
Persistent	3 (7.1%)	2 (3.6%)
Permanent	16 (38.1%)	20 (36.4%)
History of electrical cardioversion	10 (23.8%)	20 (36.4%)
History of ablation	6 (14.3%)	4 (7.3%)
Comorbidities		
Arterial hypertension	38 (90.5%)	50 (90.9%)
Diabetes mellitus type 2	14 (33.3%)	18 (32.7%)
Coronary artery disease	15 (35.7%)	17 (30.9%)
Peripheral artery disease	6 (14.3%)	3 (5.5%)
Heart failure	20 (47.6%)	13 (23.6%)
History of myocardial infarction	9 (21.4%)	7 (12.7%)
History of stroke, TIA, and systemic embolism	8 (19.0%)	15 (27.3%)
History of cancer	8 (19.0%)	14 (25.5%)
History of bleeding	9 (21.4%)	10 (18.2%)
CHA_2_DS_2_-VASc score, median (IQR)	4 (3–6)	4 (3–5)
HAS-BLED score, median (IQR)	2 (1–3)	2 (1–2)
Medication		
Anticoagulant therapy		
DOAC	18 (42.9%)	26 (47.3%)
Vitamin K-Antagonist	23 (54.8%)	26 (47.3%)
Other	0 (0%)	3 (5.5%)
None	1 (2.4%)	0 (0%)
Platelet inhibitor therapy		
Acetylsalicylic acid	10 (23.8%)	6 (10.9%)
P2Y12-Inhibitors	4 (9.5%)	3 (5.5%)

Square brackets indicate the number of missing values. Abbreviations: IQR, interquartile range; TIA, transient ischemic attack.

**Table 2 ijms-24-03861-t002:** Baseline characteristics of patients with atrial fibrillation: cases with major adverse cardiovascular events vs. matched controls without major adverse cardiovascular events.

	Cases with MACE at 1 Year (n = 37)	Controls (n = 65)
Demographics		
Age, years	75.0 (69.0–79.0)	74.0 (67.0–79.0)
Male sex	24 (64.9%)	41 (64.6%)
BMI, kg/m^2^ [3]	27.1 (23.8–29.2)	26.5 (24.7–31.3)
Information on atrial fibrillation, n (%)		
Type of atrial fibrillation [5]		
First onset	5 (13.5%)	8 (12.3%)
Paroxysmal	15 (40.5%)	26 (40.0%)
Persistent	1 (2.7%)	3 (4.6%)
Permanent	15 (40.5%)	24 (36.9%)
History of electrical cardioversion	14 (37.8%)	18 (27.7%)
History of ablation	2 (5.4%)	8 (12.3%)
Comorbidities		
Arterial hypertension	30 (81.1%)	59 (90.8%)
Diabetes mellitus type 2	14 (37.8%)	21 (32.3%)
Coronary artery disease	11 (29.7%)	21 (32.3%)
Peripheral artery disease	10 (27.0%)	5 (7.7%)
Heart failure	19 (51.4%)	18 (27.7%)
History of myocardial infarction	8 (21.6%)	8 (12.3%)
History of stroke, TIA, systemic embolism	10 (27.0%)	16 (24.6%)
History of cancer	9 (24.3%)	15 (23.1%)
History of bleeding	12 (32.4%)	13 (20.0%)
CHA_2_DS_2_-VASc score, median (IQR)	4 (3–6)	4 (3–5)
HAS-BLED score, median (IQR)	2 (1–3)	2 (1–2)
Medication		
Anticoagulant therapy		
DOAC	19 (51.4%)	35 (53.8%)
Vitamin K-Antagonist	14 (37.8%)	27 (41.5%)
Other	2 (5.4%)	1 (1.5%)
None	1 (2.7%)	2 (3.1%)
Platelet inhibitor therapy		
Acetylsalicylic acid	7 (18.9%)	10 (15.4%)
P2Y12-Inhibitors	5 (13.5%)	4 (6.2%)

Square brackets indicate the number of missing values. Abbreviations: IQR, interquartile range; MACE, major adverse cardiovascular events; and TIA, transient ischemic attack.

## Data Availability

The data presented in this study are available on request from the corresponding author.

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
