# Peer review of "Circulatory miR-411-5p as a Novel Prognostic Biomarker for Major Adverse Cardiovascular Events in Patients with Atrial Fibrillation"

_ijms, 2023, doi:10.3390/ijms24043861_

Round 1
Reviewer 1 Report
Discussion and conclusion need refinement, crosstalk of the miRNAs needs some refinement as well.
Overall good manuscript.
Reviewer 2 Report
This study aimed to investigate circulating microRNAs as prognostic biomarkers for major adverse cardiovascular events (MACE) in AF patients. The authors conducted a three-stage nested case-control study within the framework of a prospective registry including 347 AF patients. In the three-stage nested case-control study, miR-411-5p was significantly associated with adverse cardiovascular events in patients with AF. Therefore, the study proposed that if miR-411-5p 236 was detectable in the circulation, patients were at approximately two-fold increased risk for cardiovascular death and MACE. Here are my suggestions:
1. In the abstract method, isn't the number of patients 347, not 418?
2. I couldn't find a citation of Figure 1 in the text. Also, place Materials and Methods next to introduction.
3. What are the selection criteria for enrolling patients in the prospective registry? Registry characteristics can be related to the homogeneity of research subjects. Please describe the characteristics of the registry in detail.
4. What programs and methods have been implemented for random assignment and proper maching? How can authors prove that the match is good? The incidence of MACE is greatly influenced by medications (anticoagulant, antiplatelet, statin, et al) and compliance (INR value, et al) in AF patients. Nevertheless, this study does not provide any information on this.
5. Is the follow-up time between MACE/CVD death and control similar?
6. In the Materials and Methods, it is ambiguous for what reasons and criteria exactly the 7 microRNAs from the discovery phase were selected.
7. The definition of MACE is different for each subgroup analysis. Why did the authors use a different MACE definition?
8. When is the cencering date for Cox proportional hazard models? How much is the follow-up loss? How were patients with discontinued follow-up dealt with in the statistical analysis?
9. In the 347 patients with AF, were there any differences in demographic characteristics and clinical variables between the two groups according to the occurrence of MACE?
10. MACE occurred in 47 cases in 42 patients. Shouldn't MACE occurrences be counted as one event occurrence per patient?
11. In Supplemental Table 2, there were 22 deaths. The text stated that 81 people died. This seems to be because the follow-up period of study subjects differs depending on the analysis. Why didn't the authors integrate and analyze the outcome data?
12. In Figure 3, the number of patients is different. Why?
13. In the Discussion, additional rationale is required for the potential mechanism by which miR-411-5p is associated with poor prognosis in patients with AF.
Round 2
Reviewer 2 Report
Thanks.